# Connection- and Node-Sparse Deep Learning: Statistical Guarantees

## Abstract

Neural networks are becoming increasingly popular in applications, but a comprehensive mathematical understanding of their potentials and limitations is still missing. In this paper, we study the prediction accuracies of neural networks from a statistical point of view. In particular, we establish statistical guarantees for deep learning with different types of sparsity-inducing regularization. Our bounds feature a mild dependence on network widths and depths, and, therefore, support the current trend toward wide and deep networks. The tools that we use in our derivations are uncommon in deep learning and, hence, might be of additional interest.

## 1 Introduction

Sparsity reduces network complexities and, consequently, lowers the demands on memory and computation, reduces overfitting, and improves interpretability (Changpinyo et al., 2017; Han et al., 2016; Kim et al., 2016; Liu et al., 2015; Wen et al., 2016). Three common notions of sparsity are connection sparsity, which means that there is only a small number of nonzero connections between nodes, node sparsity, which means that there is only a small number of active nodes (Alvarez & Salzmann, 2016; Changpinyo et al., 2017; Feng & Simon, 2017; Kim et al., 2016; Lee et al., 2008; Liu et al., 2015; Nie et al., 2015; Scardapane et al., 2017; Wen et al., 2016), and layer sparsity, which means that there is only a small number of active layers (Hebiri & Lederer, 2020). Approaches to achieving sparsity include augmenting small networks (Ash, 1989; Bello, 1992), pruning large networks (Simonyan & Zisserman, 2015; Han et al., 2016), constraint estimation (Ledent et al., 2019; Neyshabur et al., 2015; Schmidt-Hieber, 2020), and statistical regularization (Taheri et al., 2020).

The many empirical observations of the benefits of sparsity have sparked interest in mathematical support in the form of statistical theories. But such theories are still scarce and, in any case, have severe limitations. For example, statistical guarantees for deep learning with connection-sparse regularization have been established in Taheri et al. (2020), but they do not cover node sparsity, which, in view of the removal of entire nodes, has become especially popular. Moreover, their estimator involves an additional parameter, their theory is limited to a single output node, and their results have a suboptimal dependence on the input vectors. Statistical guarantees for constraint estimation over connection- and node-sparse networks follow from combining results in Neyshabur et al. (2015) and Bartlett & Mendelson (2002). But for computational and practical reasons, regularized estimation is typically preferred over constraint estimation in deep learning as well as in machine learning at large (Hastie et al., 2015). Moreover, their theory is limited to a single output node and ReLU activation, scales exponentially in the number of layers, and requires bounded loss functions. Statistical prediction guarantees for constraint estimation over connection-sparse networks have been derived in Schmidt-Hieber (2020), but their theory is limited to a single output node and ReLU activation and assumes bounded weights. In short, the existing statistical theory for deep learning with connection and node sparsity is still deficient.

The goal of this paper is to provide an improved theory for sparse deep learning. We focus on regression-type settings with layered, feedforward neural networks. The estimators under consideration consist of a standard least-squares estimator with additional regularizers that induce connection or node sparsity. We then derive our guarantees by using techniques from high-dimensional statistics (Dalalyan et al., 2017) and empirical process theory (van de Geer, 2000). In the case of

subgaussian noise, we find the rates

$$\sqrt{\frac{l\left(\log[mn\overline{p}]\right)^3}{n}} \quad \text{and} \quad \sqrt{\frac{ml\underline{p}(\log[mn\overline{p}])^3}{n}}$$

for the connection-sparse and node-sparse estimators, respectively, where $l$ is the number of hidden layers, $m$ the number of output nodes, $n$ the number of samples, $\overline{p}$ the total number of parameters, and $\underline{p}$ the maximal width of the network. The rates suggest that sparsity-inducing approaches can provide accurate prediction even in very wide (with connection sparsity) and very deep (with either type of sparsity) networks while, at the same time, ensuring low network complexities. These findings underpin the current trend toward sparse but wide and especially deep networks from a statistical perspective.

**Outline of the paper** Section 2 recapitulates the notions of connection and node sparsity and introduces the corresponding deep learning framework and estimators. Section 3 confirms the empirically-observed accuracies of connection- and node-sparse estimation in theory. Section 4 summarizes the key features and limitations of our work. The Appendix contains all proofs.

## 2 CONNECTION- AND NODE-SPARSE DEEP LEARNING

We consider data $(\boldsymbol{y}_1, \boldsymbol{x}_1), \dots, (\boldsymbol{y}_n, \boldsymbol{x}_n) \in \mathbb{R}^m \times \mathbb{R}^d$ that are related via

$$\boldsymbol{y}_i = \boldsymbol{g}_*[\boldsymbol{x}_i] + \boldsymbol{u}_i \qquad \text{for } i \in \{1, \dots, n\} \tag{1}$$

for an unknown data-generating function $\boldsymbol{g}_* : \mathbb{R}^d \to \mathbb{R}^m$ and unknown, random noise $\boldsymbol{u}_1, \dots, \boldsymbol{u}_n \in \mathbb{R}^m$. We allow all aspects, namely $\boldsymbol{y}_i, \boldsymbol{g}_*, \boldsymbol{x}_i$, and $\boldsymbol{u}_i$, to be unbounded. Our goal is to model the data-generating function with a feedforward neural network of the form

$$\boldsymbol{g}_{\boldsymbol{\Theta}}[\boldsymbol{x}] := \Theta^l \boldsymbol{f}^l \big[\Theta^{l-1} \cdots \boldsymbol{f}^1[\Theta^0 \boldsymbol{x}]\big] \qquad \text{for } \boldsymbol{x} \in \mathbb{R}^d \tag{2}$$

indexed by the parameter space $\mathcal{M} := \{\boldsymbol{\Theta} = (\Theta^l, \dots, \Theta^0) : \Theta^j \in \mathbb{R}^{p^{j+1} \times p^j}\}$. The functions $\boldsymbol{f}^j : \mathbb{R}^{p^j} \to \mathbb{R}^{p^j}$ are called the activation functions, and $p^0 := d$ and $p^{l+1} := m$ are called the input and output dimensions, respectively. The depth of the network is $l$, the maximal width is $\underline{p} := \max_{j \in \{0, \dots, l-1\}} p^{j+1}$, and the total number of parameters is $\overline{p} := \sum_{j=0}^l p^{j+1} p^j$.

In practice, the total number of parameters often rivals or exceeds the number of samples: $\overline{p} \approx n$ or $\overline{p} \gg n$. We then speak of high dimensionality. A common technique for avoiding overfitting in high-dimensional settings is regularization that induces additional structures, such as sparsity. Sparsity has the interesting side-effect of reducing the networks' complexities, which can facilitate interpretations and reduce demands on energy and memory. Our first sparse estimator is

$$\widehat{\boldsymbol{\Theta}}_{\text{con}} \in \operatorname*{arg\,min}_{\boldsymbol{\Theta} \in \mathcal{M}_1} \left\{ \sum_{i=1}^n \big\|\boldsymbol{y}_i - \boldsymbol{g}_{\boldsymbol{\Theta}}[\boldsymbol{x}_i]\big\|_2^2 + r_{\text{con}} \|\!|\Theta^l|\!\|_1 \right\} \tag{3}$$

for a tuning parameter $r_{\text{con}} \in [0, \infty)$, a nonempty set of parameters

$$\mathcal{M}_1 \subset \left\{ \boldsymbol{\Theta} \in \mathcal{M} : \max_{j \in \{0, \dots, l-1\}} \|\!|\Theta^j|\!\|_1 \leq 1 \right\},$$

and the $\ell_1$-norm

$$\|\!|\Theta^j|\!\|_1 := \sum_{i=1}^{p^{j+1}} \sum_{k=1}^{p^j} |(\Theta^j)_{ik}| \qquad \text{for } j \in \{0, \dots, l\}, \Theta^j \in \mathbb{R}^{p^{j+1} \times p^j}.$$

This estimator is an analog of the lasso estimator in linear regression (Tibshirani, 1996). It induces sparsity on the level of connections: the larger the tuning parameter $r_{\text{con}}$, the fewer connections among the nodes.

Deep learning with $\ell_1$-regularization has become common in theory and practice (Kim et al., 2016; Taheri et al., 2020). Our estimator (3) specifies one way to formulate this type of regularization. The estimator is indeed a regularized estimator (rather than a constraint estimator), because the complexity

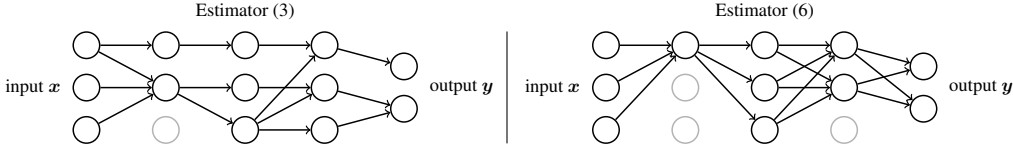

Figure 1: exemplary networks produced by the connection-sparse estimator (3) and the node-sparse estimator (6)

is regulated entirely through the tuning parameter $r_{\mathrm{con}}$ in the objective function (rather than through a tuning parameter in the set over which the objective function is optimized). But $\ell_1$-regularization could also be formulated slightly differently. For example, one could consider the estimators

$$\overline{\boldsymbol{\Theta}}_{\mathrm{con}} \in \underset{\boldsymbol{\Theta} \in \mathcal{M}}{\arg\min} \left\{ \sum_{i=1}^{n} \left\| \boldsymbol{y}_i - \boldsymbol{g}_{\boldsymbol{\Theta}}[\boldsymbol{x}_i] \right\|_2^2 + r_{\mathrm{con}} \prod_{j=0}^{l} \|\!|\Theta^j|\!\|_1 \right\} \tag{4}$$

or

$$\widetilde{\boldsymbol{\Theta}}_{\mathrm{con}} \in \underset{\boldsymbol{\Theta} \in \mathcal{M}}{\arg\min} \left\{ \sum_{i=1}^{n} \left\| \boldsymbol{y}_i - \boldsymbol{g}_{\boldsymbol{\Theta}}[\boldsymbol{x}_i] \right\|_2^2 + r_{\mathrm{con}} \sum_{j=0}^{l} \|\!|\Theta^j|\!\|_1 \right\}. \tag{5}$$

The differences among the estimators (3)–(5) are small: for example, our theory can be adjusted for (4) with almost no changes of the derivations. The differences among the estimators mainly concern the normalizations of the parameters; we illustrate this in the following proposition.

**Proposition 1** (Scaling of Norms). *Assume that the all-zeros parameter $(\mathbf{0}_{p^{l+1} \times p^l}, \ldots, \mathbf{0}_{p^1 \times p^0}) \in \mathcal{M}_1$ is neither a solution of (3) nor of (5), that $r_{\mathrm{con}} > 0$, and that the activation functions are nonnegative homogenous: $\boldsymbol{f}^j[a\boldsymbol{b}] = a\boldsymbol{f}^j[\boldsymbol{b}]$ for all $j \in \{1, \ldots, l\}$, $a \in [0, \infty)$, and $\boldsymbol{b} \in \mathbb{R}^{p^j}$. Then, $\|\!|(\widehat{\Theta}_{\mathrm{con}})^0|\!\|_1, \ldots, \|\!|(\widehat{\Theta}_{\mathrm{con}})^{l-1}|\!\|_1 = 1$ (concerns the inner layers) for all solutions of (3), while $\|\!|(\widetilde{\Theta}_{\mathrm{con}})^0|\!\|_1 = \cdots = \|\!|(\widetilde{\Theta}_{\mathrm{con}})^l|\!\|_1$ (concerns all layers) for at least one solution of (5).*

Another way to formulate $\ell_1$-regularization was proposed in Taheri et al. (2020): they reparametrize the networks through a scale parameter and a constraint version of $\mathcal{M}$ and then to focus the regularization on the scale parameter only. Our above-stated estimator (3) is more elegant in that it avoids the reparametrization and the additional parameter.

The factor $\|\!|\Theta^l|\!\|_1$ in the regularization term of (3) measures the complexity of the network over the set $\mathcal{M}_1$, and the factor $r_{\mathrm{con}}$ regulates the complexity of the resulting estimator. This provides a convenient lever for data-adaptive complexity regularization through well-established calibration schemes for the tuning parameter, such as cross-validation. This practical aspect is an advantage of regularized formulations like ours as compared to constraint estimation over sets with a predefined complexity.

The constraints in the set $\mathcal{M}_1$ of the estimator (3) can also retain the expressiveness of the full parameterization that corresponds to the set $\mathcal{M}$: for example, assuming again nonnegative-homogeneous activation, one can check that for every $\boldsymbol{\Gamma} \in \mathcal{M}$, there is a $\boldsymbol{\Gamma}' \in \{\boldsymbol{\Theta} \in \mathcal{M} : \max_{j \in \{0, \ldots, l-1\}} \|\!|\Theta^j|\!\|_1 \leq 1\}$ such that $\boldsymbol{g}_{\boldsymbol{\Gamma}} = \boldsymbol{g}_{\boldsymbol{\Gamma}'}$—cf. Taheri et al. (2020, Proposition 1). In contrast, existing theories on neural networks often require the parameter space to be bounded, which limits the expressiveness of the networks.

Our regularization approach is, therefore, closer to practical setups than constraint approaches. The price is that to develop prediction theories, we have to use different tools than those typically used in theoretical deep learning. For example, we cannot use established risk bounds such as Bartlett & Mendelson (2002, Theorem 8) (because Rademacher complexities over classes of unbounded functions are unbounded) or Lederer (2020a, Theorem 1) (because our loss function is not Lipschitz continuous) or established concentration bounds such as McDiarmid's inequality in McDiarmid (1989, Lemma (3.3)) (because that would require a bounded loss). We instead invoke ideas from high-dimensional statistics, prove Lipschitz properties for neural networks, and use empirical process theory that is based on chaining (see the Appendix).

Our second estimator is

$$\widehat{\boldsymbol{\Theta}}_{\text{node}} \in \underset{\boldsymbol{\Theta} \in \mathcal{M}_{2,1}}{\arg\min} \left\{ \sum_{i=1}^{n} \left\| \boldsymbol{y}_i - \boldsymbol{g}_{\boldsymbol{\Theta}}[\boldsymbol{x}_i] \right\|_2^2 + r_{\text{node}} \|\!|\Theta^l|\!\|_{2,1} \right\} \tag{6}$$

for a tuning parameter $r_{\text{node}} \in [0, \infty)$, a nonempty set of parameters

$$\mathcal{M}_{2,1} \subset \left\{ \boldsymbol{\Theta} \in \mathcal{M} \; : \; \max_{j \in \{0, \ldots, l-1\}} \|\!|\Theta^j|\!\|_{2,1} \leq 1 \right\},$$

and the $\ell_2/\ell_1$-norm

$$\|\!|\Theta^j|\!\|_{2,1} := \sum_{k=1}^{p^j} \sqrt{\sum_{i=1}^{p^{j+1}} |(\Theta^j)_{ik}|^2} \qquad \text{for } j \in \{0, \ldots, l-1\}, \, \Theta^j \in \mathbb{R}^{p^{j+1} \times p^j} \,.$$

This estimator is an analog of the group-lasso estimator in linear regression (Bakin, 1999). Again, to avoid ambiguities in the regularization, our formulation is slightly different from the standard formulations in the literature, but the fact that group-lasso regularizers leads to node-sparse networks has been discussed extensively before (Alvarez & Salzmann, 2016; Liu et al., 2015; Scardapane et al., 2017): the larger the tuning parameter $r_{\text{node}}$, the fewer active nodes in the network.

The above-stated comments about the specific form of the connection-sparse estimator also apply to the node-sparse estimator.

An illustration of connection and node sparsity is given in Figure 1. Connection-sparse networks have only a small number of active connections between nodes (left panel of Figure 1); node-sparse networks have inactive nodes, that is, completely unconnected nodes (right panel of Figure 1). The two notions of sparsity are connected: for example, connection sparsity can render entire nodes inactive "by accident" (see the layer that follows the input layer in the left panel of the figure). In general, node sparsity is the weaker assumption, because it allows for highly connected nodes; this observation is reflected in the theoretical guarantees in the following section.

The optimal network architecture for given data (such as the optimal width) is hardly known beforehand in a data analysis. A main feature of sparsity-inducing regularization is, therefore, that it adjusts parts of the network architecture to the data. In other words, sparsity-inducing regularization is a data-driven approach to adapting the complexity of the network.

While versions of the estimators (3) and (6) are popular in deep learning, statistical analyses, especially of node-sparse deep learning, are scarce. Such a statistical analysis is, therefore, the goal of the following section.

## 3 STATISTICAL PREDICTION GUARANTEES

We now develop statistical guarantees for the sparse estimators described above. The guarantees are formulated in terms of the squared *average (in-sample) prediction error*

$$\text{err}[\boldsymbol{\Theta}] := \frac{1}{n} \sum_{i=1}^{n} \left\| \boldsymbol{g}_*[\boldsymbol{x}_i] - \boldsymbol{g}_{\boldsymbol{\Theta}}[\boldsymbol{x}_i] \right\|_2^2 \quad \text{for } \boldsymbol{\Theta} \in \mathcal{M} \,,$$

which is a measure for how well the network $\boldsymbol{g}_{\boldsymbol{\Theta}}$ fits the unknown function $\boldsymbol{g}_*$ (which does not need to be a neural network) on the data at hand, and in terms of the *prediction risk* (or *generalization error*) for a new sample $(\boldsymbol{y}, \boldsymbol{x})$ that has the same distribution as the original data

$$\text{risk}[\boldsymbol{\Theta}] := E\|\boldsymbol{y} - \boldsymbol{g}_{\boldsymbol{\Theta}}[\boldsymbol{x}]\|_2^2 \quad \text{for } \boldsymbol{\Theta} \in \mathcal{M} \,,$$

which measures how well the network $\boldsymbol{g}_{\boldsymbol{\Theta}}$ can predict a new sample. We first study the prediction error, because it is agnostic to the distribution of the input data; in the end, we then translate the bounds for the prediction error into bounds for the generalization error.

We first observe that the networks in (2) can be somewhat "linearized:" For every parameter $\boldsymbol{\Theta} \in \mathcal{M}_1$, there is a parameter

$$\overline{\boldsymbol{\Theta}} \in \overline{\mathcal{M}}_1 := \left\{ \overline{\boldsymbol{\Theta}} = (\overline{\Theta}^{l-1}, \ldots, \overline{\Theta}^0) \; : \; \overline{\Theta}^j \in \mathbb{R}^{p^{j+1} \times p^j}, \, \max_{j \in \{0, \ldots, l-1\}} \|\!|\overline{\Theta}^j|\!\|_1 \leq 1 \right\}$$

such that for every $x \in \mathbb{R}^d$

$$g_\Theta[x] = \Theta^l \overline{g}_{\overline{\Theta}}[x] \quad \text{with} \quad \overline{g}_{\overline{\Theta}}[x] := f^l[\overline{\Theta}^{l-1} \cdots f^1[\overline{\Theta}^0 x]] \in \mathbb{R}^{p^l}. \tag{7}$$

This additional notation allows us to disentangle the outermost layer (which is regularized directly) from the other layers (which are regularized indirectly). More generally speaking, the additional notation makes a connection to linear regression, where the above holds trivially with $\overline{g}_{\overline{\Theta}}[x] = x$.

We also define

$$\overline{\mathcal{M}}_{2,1} := \left\{ \overline{\Theta} = (\overline{\Theta}^{l-1}, \ldots, \overline{\Theta}^0) \,:\, \overline{\Theta}^j \in \mathbb{R}^{p^{j+1} \times p^j}, \max_{j \in \{0, \ldots, l-1\}} \|\overline{\Theta}^j\|_{2,1} \le 1 \right\}$$

accordingly.

In high-dimensional linear regression, the quantity central to prediction guarantees is the *effective noise* (Lederer & Vogt, 2020). The effective noise is in our notation (with $l = 0$ and $m = 1$ to describe linear regression) $2\|\sum_{i=1}^n u_i x_i\|_\infty$. The above linearization allows us to generalize the effective noise to our general deep-learning framework:

$$
\begin{aligned}
r^*_{\text{con}} &:= 2 \sup_{\overline{\Psi} \in \overline{\mathcal{M}}_1} \left\| \sum_{i=1}^n u_i (\overline{g}_{\overline{\Psi}}[x_i])^\top \right\|_\infty \\
r^*_{\text{node}} &:= 2\sqrt{m} \sup_{\overline{\Psi} \in \overline{\mathcal{M}}_{2,1}} \left\| \sum_{i=1}^n u_i (\overline{g}_{\overline{\Psi}}[x_i])^\top \right\|_\infty,
\end{aligned}
\tag{8}
$$

where $\|A\|_\infty := \max_{(i,j) \in \{1,\ldots,m\} \times \{1,\ldots,p^l\}} |A_{ij}|$ for $A \in \mathbb{R}^{m \times p^l}$. The effective noises, as we will see below, are the optimal tuning parameters in our theories; at the same time, the effective noises depend on the noise random variables $u_1, \ldots, u_n$, which are unknown in practice. Accordingly, we call the quantities $r^*_{\text{con}}$ and $r^*_{\text{node}}$ the *oracle tuning parameters*.

We take a moment to compare the effective noises in (8) to Rademacher complexities (Koltchinskii, 2001; Koltchinskii & Panchenko, 2002). Rademacher complexities are the basis of a line of other statistical theories for deep learning (Bartlett & Mendelson, 2002; Golowich et al., 2017; Lederer, 2020a; Neyshabur et al., 2015). In our framework, the Rademacher complexities in the case $m = 1$ are (Lederer, 2020a, Definition 1)

$$\mathbb{E}_{x_1, \ldots, x_n, k_1, \ldots, k_n} \left[ \sup_{\Theta \in \mathcal{M}_1} \left| \frac{1}{n} \sum_{i=1}^n k_i g_\Theta[x_i] \right| \right] \quad \text{and} \quad \mathbb{E}_{x_1, \ldots, x_n, k_1, \ldots, k_n} \left[ \sup_{\Theta \in \mathcal{M}_{2,1}} \left| \frac{1}{n} \sum_{i=1}^n k_i g_\Theta[x_i] \right| \right]$$

for i.i.d. Rademacher random variables $k_1, \ldots, k_n$. The effective noises might look like (rescaled) empirical versions of these quantities at first sight, but this is not the case. Two immediate differences are that (8) apply to general $m$ and circumvent the outermost layers of the networks. But more importantly, Rademacher complexities involve external i.i.d. Rademacher random variables that are not connected with the statistical model at hand, while the effective noises involve the noise variables, which are completely specified by the model and, therefore, can have any distribution (see our sub-Gaussian example further below). Hence, there are no general techniques to relate Rademacher complexities and effective noises.

Not only are the two concepts distinct, but also they are used in very different ways. For example, existing theories use Rademacher complexities to measure the size of the function class at hand, while we use effective noises to measure the maximal impact of the stochastic noise on the estimators. (Our proofs also require a measure of the size of the function class, but this measure is entropy—cf. Lemma 1.) In general, our proof techniques are very different from those in the context of Rademacher complexities.

We can now state a general prediction guarantee.

**Theorem 1** (General Prediction Guarantees). *If $r_{\text{con}} \ge r^*_{\text{con}}$, it holds that*

$$\text{err}[\widehat{\Theta}_{\text{con}}] \le \inf_{\Theta \in \mathcal{M}_1} \left\{ \text{err}[\Theta] + \frac{2r_{\text{con}}}{n} \|\Theta^l\|_1 \right\}.$$

*Similarly, if $r_{\text{node}} \ge r^*_{\text{node}}$, it holds that*

$$\text{err}[\widehat{\Theta}_{\text{node}}] \le \inf_{\Theta \in \mathcal{M}_{2,1}} \left\{ \text{err}[\Theta] + \frac{2r_{\text{node}}}{n} \|\Theta^l\|_{2,1} \right\}.$$

Each bound contains an approximation error $\mathrm{err}[\boldsymbol{\Theta}]$ that captures how well the class of networks can approximate the true data-generating function $\boldsymbol{g}_*$ and a statistical error proportional to $r_{\mathrm{con}}/n$ and $r_{\mathrm{node}}/n$, respectively, that captures how well the estimator can select within the class of networks at hand. In other words, Theorem 1 ensures that the estimators (3) and (6) predict—up to the statistical error described by $r_{\mathrm{con}}/n$ and $r_{\mathrm{node}}/n$, respectively—as well as the best connection- and node-sparse network. This observation can be illustrated further:

**Corollary 1** (Parametric Setting). *If additionally $\boldsymbol{g}_* = \boldsymbol{g}_{\boldsymbol{\Theta}^*}$ for a $\boldsymbol{\Theta}^* \in \mathcal{M}_1$, it holds that*

$$\mathrm{err}[\widehat{\boldsymbol{\Theta}}_{\mathrm{con}}] \leq \frac{2r_{\mathrm{con}}}{n} \|\!|(\boldsymbol{\Theta}^*)^l|\!\|_1 \, .$$

*If instead $\boldsymbol{g}_* = \boldsymbol{g}_{\boldsymbol{\Theta}^*}$ for a $\boldsymbol{\Theta}^* \in \mathcal{M}_{2,1}$, it holds that*

$$\mathrm{err}[\widehat{\boldsymbol{\Theta}}_{\mathrm{node}}] \leq \frac{2r_{\mathrm{node}}}{n} \|\!|(\boldsymbol{\Theta}^*)^l|\!\|_{2,1} \, .$$

Hence, if the underlying data-generating function is a sparse network itself, the prediction errors of the estimators are essentially bounded by the statistical errors $r_{\mathrm{con}}/n$ and $r_{\mathrm{node}}/n$.

The above-stated results also identify the oracle tuning parameters $r_{\mathrm{con}}^*$ and $r_{\mathrm{node}}^*$ as optimal tuning parameters: they give the best prediction guarantees in Theorem 1. But since the oracle tuning parameters are unknown in practice, the guarantees implicitly presume a calibration scheme that satisfies $r_{\mathrm{con}} \approx r_{\mathrm{con}}^*$ in practice. A natural candidate is cross-validation, but there are no guarantees that cross-validation provides such tuning parameters. This is a limitation that our theories share with all other theories in the field.

Rather than dealing with the practical calibration of the tuning parameters, we exemplify the oracle tuning parameters in a specific setting. This analysis will illustrate the rates of convergences that we can expect from Theorem 1, and it will allow us to compare our theories with other theories in the literature. Assume that the activation functions satisfy $\boldsymbol{f}^j[\boldsymbol{0}_{p^j}] = \boldsymbol{0}_{p^j}$ and are 1-Lipschitz continuous with respect to the Euclidean norms on the functions' input and output spaces $\mathbb{R}^{p^j}$. A popular example is ReLU activation (Nair & Hinton, 2010), but the conditions are met by many other functions as well. Also, assume that the noise vectors $\boldsymbol{u}_1, \ldots, \boldsymbol{u}_n$ are independent and centered and have uniformly subgaussian entries (van de Geer, 2000, Display (8.2) on Page 126). Keep the input vectors fixed and capture their normalizations by

$$\overline{v}_\infty := \sqrt{\frac{1}{n} \sum_{i=1}^n \|\boldsymbol{x}_i\|_\infty^2} \quad \text{and} \quad \overline{v}_2 := \sqrt{\frac{1}{n} \sum_{i=1}^n \|\boldsymbol{x}_i\|_2^2} \, .$$

Then, we obtain the following bounds for the effective noises.

**Proposition 2** (Subgaussian Noise). *There is a constant $c \in (0, \infty)$ that depends only on the subgaussian parameters of the noise such that*

$$P\left\{ r_{\mathrm{con}}^* \leq c\overline{v}_\infty \sqrt{nl\big(\log[2mn\overline{p}]\big)^3} \right\} \geq 1 - \frac{1}{n}$$

*and*

$$P\left\{ r_{\mathrm{node}}^* \leq c\overline{v}_2 \sqrt{mnl\underline{p}\big(\log[2mn\overline{p}]\big)^3} \right\} \geq 1 - \frac{1}{n} \, .$$

Broadly speaking, this result combined with Theorem 1 illustrates that accurate prediction with connection- and node-sparse estimators is possible even when using very wide and deep networks. Let us analyze the factors one by one and compare them to the factors in the bounds of Taheri et al. (2020) and Neyshabur et al. (2015), which are the two most related papers. The connection-sparse case compares to the results in Taheri et al. (2020), and it compares to the results in Neyshabur et al. (2015) when setting the parameters in that paper to $p = q = 1$ (which gives a setting that is slightly more restrictive than ours) or $p = 1; q = \infty$ (which gives a setting that is slightly less restrictive than ours). The node-sparse case compares to Neyshabur et al. (2015) with $p = 2; q = \infty$ (which gives a setting that is more restrictive than ours, though). Our setup is also more general than the one in Neyshabur et al. (2015) in the sense that it allows for activation other than ReLU.

The dependence on $n$ is, as usual, $1/\sqrt{n}$ up to logarithmic factors.

In the connection-sparse case, our bounds involve $\overline{v}_\infty = \sqrt{\sum_{i=1}^n \|x_i\|_\infty^2/n}$ rather than the factor $v_\infty := \max_{i\in\{1,\dots,n\}} \|x_i\|_\infty$ of Neyshabur et al. (2015) or the factor $\overline{v}_2 = \sqrt{\sum_{i=1}^n \|x_i\|_2^2/n}$ of Taheri et al. (2020). In principle, the improvements of $\overline{v}_\infty$ over $v_\infty$ and $\overline{v}_2$ can be up to a factor $\sqrt{n}$ and up to a factor $\sqrt{d}$, respectively; in practice, the improvements depend on the specifics on the data. For example, on the training data of `MNIST` (LeCun et al., 1998) and `Fashion-MNIST` (Xiao et al., 2017) ($\sqrt{n} \approx 250$; $\sqrt{d} = 28$ in both data sets), it holds that $\overline{v}_\infty \approx v_\infty \approx \overline{v}_2/9$ and $\overline{v}_\infty \approx v_\infty \approx \overline{v}_2/12$, respectively. In the node-sparse case, our bounds involve $\overline{v}_2$, which is again somewhat smaller than the factor $v_2 := \max_{i\in\{1,\dots,n\}} \|x_i\|_2$ in Neyshabur et al. (2015).

The main difference between the bounds for the connection-sparse and node-sparse estimators are their dependencies on the networks' maximal width $p$. The bound for the connection-sparse estimator (3) depends on the width $p$ only logarithmically (through $\overline{p}$), while the bound for the node-sparse estimator (6) depends on $\overline{p}$ sublinearly. The dependence in the connection-sparse case is the same as in Taheri et al. (2020), while Neyshabur et al. (2015) can avoid even that logarithmic dependence (and, therefore, allow for networks with infinite widths). The node-sparse case in Neyshabur et al. (2015) does not involve our linear dependence on the width, but this difference stems from the fact that they use a more restrictive version of the grouping—we take the maximum over each layer, while they take the maximum over each node— and our results can be readily adjusted to their notion of group sparsity. These observations indicate that node sparsity as formulated above is suitable for slim networks ($p \ll n$) but should be strengthened or complemented with other notions of sparsity otherwise. To give a numeric example, the training data in `MNIST` (LeCun et al., 1998) and `Fashion-MNIST` (Xiao et al., 2017) comprise $n = 60\,000$ samples, which means that the width should be considerably smaller than $60\,000$ when using node sparsity alone. (Note that the input layer does not take part in $p$, which means that $d$ could be larger.)

For unconstraint estimation, one can expect a linear dependence of the error on the total number of parameters (Anthony & Bartlett, 1999). Our bounds for the sparse estimators, in contrast, only have a $\log[\overline{p}]$ dependence on the total number of parameters. This difference illustrates the virtue of regularization in general, and the virtue of sparsity in particular.

Both of our bounds have a mild $\sqrt{l}$ dependence on the depth. These dependencies considerably improve on the exponentially-increasing dependencies on the depth in Neyshabur et al. (2015) and, therefore, are particularly suited to describe deep network architectures. Replacing the conditions $\max_j \|\Theta^j\|_1 \le 1$ and $\max_j \|\Theta^j\|_{2,1} \le 1$ in the definitions of the connection-sparse and node-sparse estimators by the stricter conditions $\sum_j \|\Theta^j\|_1 \le 1$ and $\sum_j \|\Theta^j\|_{2,1} \le 1$, respectively (cf. Taheri et al. (2020) and our discussion in Section 2), the dependence on the depth can be improved further from $\sqrt{l}$ to $(2/l)^l \sqrt{l}$ (this only requires a simple adjustment of the last display in the proof of Proposition 4), which is exponentially decreasing in the depth.

Our connection-sparse bounds have a mild $\log[m]$ dependence on the number of output nodes; the node-sparse bound involve an additional factor $\sqrt{m}$. The case of multiple outputs has not been considered in statistical prediction bounds before.

Proposition 2 also highlights another advantage of our regularization approach over theories such as Neyshabur et al. (2015) that apply to constraint estimators. The theories for constraint estimators require bounding the sparsity levels directly, but in practice, suitable values for these bounds are rarely known. In our framework, in contrast, the sparsity is controlled via tuning parameters indirectly, and Proposition 2—although not providing a complete practical calibration scheme—gives insights into how these tuning parameters should scale with $n$, $d$, $l$, and so forth.

We also note that the bounds in Theorem 1 can be generalized readily to every estimator of the form

$$\widehat{\Theta}_{\text{gen}} \in \underset{\Theta\in\mathcal{M}_{\text{gen}}}{\arg\min}\left\{\sum_{i=1}^n \left\|y_i - g_\Theta[x_i]\right\|_2^2 + r_{\text{gen}}\|\Theta^l\|\right\},$$

where $r_{\text{gen}} \in [0,\infty)$ is a tuning parameter, $\mathcal{M}_{\text{gen}}$ any nonempty subset of $\mathcal{M}$, and $\|\cdot\|$ any norm. The bound for such an estimator is then

$$\text{err}[\widehat{\Theta}_{\text{gen}}] \le \inf_{\Theta\in\mathcal{M}_{\text{gen}}}\left\{\text{err}[\Theta] + \frac{2r_{\text{gen}}}{n}\|\Theta^l\|\right\}$$

| Approach | Basis | References |
|---|---|---|
| FatShat | Fat-shattering dimension | Bartlett (1998) |
| Rad1 | Rademacher complexity | Bartlett & Mendelson (2002); Lederer (2020a); Neyshabur et al. (2015) |
| Rad2 | Rademacher complexity | Bartlett & Mendelson (2002); Golowich et al. (2017); Lederer (2020a) |
| NonPar | Non-parametric statistics | Schmidt-Hieber (2020) |
| HighDim | High-dimensional statistics | Taheri et al. (2020) |

| Approach | Feasible | $\ell_2$-loss | Regularized | (sub)linear in $l$ | node sparsity | multiple outputs |
|---|---|---|---|---|---|---|
| FatShat | ✓ | | | | | |
| Rad1 | ✓ | | | | ✓ | |
| Rad2 | ✓ | | | ✓ | | |
| NonPar | | ✓ | | ✓ | | |
| HighDim | ✓ | ✓ | ✓ | ✓ | | |

Table 1: presence (✓) or absence (  ) of certain features in previous statistical theories for sparse deep learning

for $r_{\text{gen}} \geq r_{\text{gen}}^*$, where $r_{\text{gen}}^*$ is as $r_{\text{con}}^*$ but based on the dual norm of $\|\cdot\|$ instead of the dual norm of $\|\cdot\|_1$. For example, one could impose connection sparsity on some layers and node sparsity on others, or one could impose different regularizations altogether. We omit the details to avoid digression.

We finally illustrate that the bounds for the prediction errors also entail bounds for the generalization errors. For simplicity, we consider a parametric setting and subgaussian noise again.

**Proposition 3** (Generalization Guarantees). *Assume that the inputs $\boldsymbol{x}, \boldsymbol{x}_1, \ldots, \boldsymbol{x}_n$ are i.i.d. random vectors, that the noise vectors $\boldsymbol{u}_1, \ldots, \boldsymbol{u}_n$ are independent and centered and have uniformly subgaussian entries, and that $r_{\text{con}}^*, r_{\text{node}}^* \to 0$ as $n \to \infty$. Consider an arbitrary positive constant $b \in (0, \infty)$. If $\boldsymbol{g}_* = \boldsymbol{g}_{\boldsymbol{\Theta}^*}$ for a $\boldsymbol{\Theta}^* \in \mathcal{M}_1$ that is independent of the sample size $n$, it holds with probability at least $1 - 1/n$ that*

$$\text{risk}[\widehat{\boldsymbol{\Theta}}_{\text{con}}] \leq (1+b)\,\text{risk}[\boldsymbol{\Theta}^*] + c\overline{v}_\infty \sqrt{\frac{l\big(\log[2mn\overline{p}]\big)^3}{n}} \,\|(\boldsymbol{\Theta}^*)^l\|_1$$

*for a constant $c \in (0, \infty)$ that depends only on $b$ and the subgaussian parameters of the noise. Similarly, if $\boldsymbol{g}_* = \boldsymbol{g}_{\boldsymbol{\Theta}^*}$ for a $\boldsymbol{\Theta}^* \in \mathcal{M}_{2,1}$ that is independent of the sample size $n$, it holds with probability at least $1 - 1/n$ that*

$$\text{risk}[\widehat{\boldsymbol{\Theta}}_{\text{con}}] \leq (1+b)\,\text{risk}[\boldsymbol{\Theta}^*] + c\overline{v}_2 \sqrt{\frac{ml\underline{p}\big(\log[2mn\overline{p}]\big)^3}{n}} \,\|(\boldsymbol{\Theta}^*)^l\|_{2,1}$$

*for a constant $c \in (0, \infty)$ that depends only on $b$ and the subgaussian parameters of the noise.*

Hence, the generalization errors are bounded by the same terms as the prediction errors.

## 4 Discussion

Our statistical theory for sparse deep learning incorporates node sparsity as well as connection sparsity, scales favorably in the number of layers, provides insights into how the tuning parameters should scale with the dimensions of the problem, and applies to unbounded loss functions. It is the first statistical theory that has all of these features—cf. Table 1. Additionally we avoid the introduction of an additional scaling parameter and improve the dependence of the rates on the input data. Finally, our novel proof approach based on high-dimensional statistics and empirical-process theory is of independent interest.

Evidence for the benefits of deep networks has been established in practice (LeCun et al., 2015; Schmidhuber, 2015), approximation theory (Liang & Srikant, 2016; Telgarsky, 2016; Yarotsky, 2017), and statistics (Golowich et al., 2017; Taheri et al., 2020). Since our guarantees scale at most sublinearly in the number of layers (or even improve with increasing depth—see our comment on Page 7), our paper complements these lines of research and shows that sparsity-inducing regularization is an effective approach to coping with the complexity of deep and very deep networks.

Connection sparsity limits the number of nonzero entries in each parameter matrix, while layer sparsity only limits the total number of nonzero rows. Hence, the number of columns in a parameter matrix, that is, the width of the preceding layer, is regularized only in the case of connection sparsity. Our theoretical results reflect this insight in that the bounds for the connection- and node-sparse estimators depend on the networks' width logarithmically and sublinearly, respectively. Practically speaking, our results indicate that connection sparsity is suitable to handle wide networks, but node sparsity is suitable only when complemented by connection sparsity or other strategies.

The mild logarithmic dependence of our connection-sparse bounds on the number of output nodes illustrates that networks with many outputs can be learned in practice. Our prediction theory is the first one that considers multiple output nodes; a classification theory with a logarithmic dependence on the output nodes has been established very recently in Ledent et al. (2019).

The mathematical underpinnings of our theory are very different from those of most other papers in theoretical deep learning. The proof of the main theorem shares similarities with proofs in high-dimensional statistics; to formulate and control the relevant empirical processes, we use the concept of effective noise, chaining, and Lipschitz properties of neural networks. These tools are not standard in deep learning theory and, therefore, might be of more general interest (see Appendix A.7 for further details).

Our theory shares some limitations with all other current theories in deep learning: the network architectures are simpler than the ones typically used in practice (cf. Lederer (2020b), though); the bounds concern global optima rather than the local optima or saddle points provided by many practical algorithms; and the theory does not entail a practical scheme for the calibration of the tuning parameters. Nevertheless, our theory, and mathematical theory in general, provides insights about what accuracies to expect in practice and about what network types and estimators might be suitable for a given problem.

In summary, our paper highlights the benefits of sparsity in deep learning and, more generally, showcases the usefulness of statistical analyses for understanding neural networks.

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

## A APPENDIX

The Appendix consists of two auxiliary results and the proofs of Theorem 1 and Propositions 1 and 2. Our approach combines techniques from high-dimensional statistics and empirical-process theory that are very different from the techniques used in most other approaches in the literature.

### A.1 LIPSCHITZ PROPERTY

In this section, we prove a Lipschitz property that we use in the proof of Proposition 2.

**Proposition 4** (Lipschitz Property). *In the framework of Sections 2 and 3, it holds for all $\overline{\boldsymbol{\Theta}}, \overline{\boldsymbol{\Gamma}} \in \overline{\mathcal{M}}_1$ that*

$$\left\|\overline{\boldsymbol{g}}_{\overline{\boldsymbol{\Theta}}}[\boldsymbol{x}] - \overline{\boldsymbol{g}}_{\overline{\boldsymbol{\Gamma}}}[\boldsymbol{x}]\right\|_\infty \le \sqrt{l}\|\boldsymbol{x}\|_\infty \|\overline{\boldsymbol{\Theta}} - \overline{\boldsymbol{\Gamma}}\|_{\mathrm{F}}$$

*and for all $\overline{\boldsymbol{\Theta}}, \overline{\boldsymbol{\Gamma}} \in \overline{\mathcal{M}}_{2,1}$ that*

$$\left\|\overline{\boldsymbol{g}}_{\overline{\boldsymbol{\Theta}}}[\boldsymbol{x}] - \overline{\boldsymbol{g}}_{\overline{\boldsymbol{\Gamma}}}[\boldsymbol{x}]\right\|_2 \le \sqrt{l}\|\boldsymbol{x}\|_2 \|\overline{\boldsymbol{\Theta}} - \overline{\boldsymbol{\Gamma}}\|_{\mathrm{F}} .$$

The Frobenius norm is defined as

$$\|\overline{\boldsymbol{\Theta}}\|_{\mathrm{F}} := \sqrt{\sum_{j=0}^{l-1} \|\overline{\Theta}^j\|_{\mathrm{F}}^2} := \sqrt{\sum_{j=0}^{l-1} \sum_{i=1}^{p^{j+1}} \sum_{k=1}^{p^j} |(\overline{\Theta}^j)_{ik}|^2} \qquad \text{for } \overline{\boldsymbol{\Theta}} \in \overline{\mathcal{M}}_{2,1} = \overline{\mathcal{M}}_1 \cup \overline{\mathcal{M}}_{2,1} .$$

Proposition 4 generalizes (Taheri et al., 2020, Proposition 2) to vector-valued network outputs and to node sparsity, and it replaces their $\|\boldsymbol{x}\|_2$ with the smaller $\|\boldsymbol{x}\|_\infty$ in the connection-sparse case.

*Proof of Proposition 4.* This proof generalizes and sharpens the proof of Taheri et al. (2020), and it simplifies some arguments of that proof. We define the "inner subnetworks" of a network $\overline{\boldsymbol{g}}_{\overline{\boldsymbol{\Theta}}}$ with $\overline{\boldsymbol{\Theta}} \in \overline{\mathcal{M}}_{2,1}$ as the vector-valued functions

$$S_0 \overline{\boldsymbol{g}}_{\overline{\boldsymbol{\Theta}}} \; : \; \mathbb{R}^d \to \mathbb{R}^{p^1}$$
$$\boldsymbol{x} \mapsto S_0 \overline{\boldsymbol{g}}_{\overline{\boldsymbol{\Theta}}}[\boldsymbol{x}] := \overline{\Theta}^0 \boldsymbol{x}$$

and

$$S_j \overline{\boldsymbol{g}}_{\overline{\boldsymbol{\Theta}}} \; : \; \mathbb{R}^d \to \mathbb{R}^{p^{j+1}}$$
$$\boldsymbol{x} \mapsto S_j \overline{\boldsymbol{g}}_{\overline{\boldsymbol{\Theta}}}[\boldsymbol{x}] := \overline{\Theta}^j \boldsymbol{f}^j \big[ \cdots \boldsymbol{f}^1 [\overline{\Theta}^0 \boldsymbol{x}] \big]$$

for $j \in \{1, \ldots, l-1\}$. Similarly, we define the "outer subnetworks" of $\overline{\boldsymbol{g}}_{\overline{\boldsymbol{\Theta}}}$ as the real-valued functions

$$S^j \overline{\boldsymbol{g}}_{\overline{\boldsymbol{\Theta}}} \; : \; \mathbb{R}^{p^j} \to \mathbb{R}^{p^l}$$
$$\boldsymbol{z} \mapsto S^j \overline{\boldsymbol{g}}_{\overline{\boldsymbol{\Theta}}}[\boldsymbol{z}] := \boldsymbol{f}^l \big[ \overline{\Theta}^{l-1} \cdots \boldsymbol{f}^j [\boldsymbol{z}] \big]$$

for $j \in \{1, \ldots, l-1\}$ and

$$S^l \overline{\boldsymbol{g}}_{\overline{\boldsymbol{\Theta}}} \; : \; \mathbb{R}^{p^l} \to \mathbb{R}^{p^l}$$
$$\boldsymbol{z} \mapsto S^l \overline{\boldsymbol{g}}_{\overline{\boldsymbol{\Theta}}}[\boldsymbol{z}] := \boldsymbol{f}^l [\boldsymbol{z}] .$$

The initial network can be split into an inner and an outer network along every layer $j \in \{1, \ldots, l\}$:

$$\overline{\boldsymbol{g}}_{\overline{\boldsymbol{\Theta}}}[\boldsymbol{x}] = S^j \overline{\boldsymbol{g}}_{\overline{\boldsymbol{\Theta}}} \big[ S_{j-1} \overline{\boldsymbol{g}}_{\overline{\boldsymbol{\Theta}}}[\boldsymbol{x}] \big] \qquad \text{for } \boldsymbol{x} \in \mathbb{R}^d .$$

We call this our *splitting argument*.

To exploit the splitting argument, we derive a contraction result for the inner subnetworks and a Lipschitz result for the outer subnetworks. We denote the $\ell_2$-operator norm of a matrix $A$, that is, the largest singular value of $A$, by $\|A\|_{\mathrm{op}}$. Using then the assumptions that the activation functions are 1-Lipschitz and $\boldsymbol{f}^j[\boldsymbol{0}_{p^j}] = \boldsymbol{0}_{p^j}$, we get for every $\overline{\boldsymbol{\Theta}} = (\overline{\Theta}^{l-1}, \ldots, \overline{\Theta}^0) \in \overline{\mathcal{M}}_{2,1}$ and $\boldsymbol{x} \in \mathbb{R}^d$ that

$$\left\| S_{j-2} \overline{\boldsymbol{g}}_{\overline{\boldsymbol{\Theta}}}[\boldsymbol{x}] \right\|_2 = \left\| \overline{\Theta}^{j-2} \boldsymbol{f}^{j-2} \big[ S_{j-3} \overline{\boldsymbol{g}}_{\overline{\boldsymbol{\Theta}}}[\boldsymbol{x}] \big] \right\|_2$$

$$\leq \big\|\overline{\Theta}^{j-2}\big\|_{\mathrm{op}}\big\|\boldsymbol{f}^{j-2}\big[S_{j-3}\overline{\boldsymbol{g}}_{\overline{\Theta}}[\boldsymbol{x}]\big]\big\|_2$$
$$\leq \big\|\overline{\Theta}^{j-2}\big\|_{\mathrm{op}}\big\|S_{j-3}\overline{\boldsymbol{g}}_{\overline{\Theta}}[\boldsymbol{x}]\big\|_2$$
$$\leq \cdots$$
$$\leq \bigg(\prod_{k=1}^{j-2}\big\|\overline{\Theta}^{k}\big\|_{\mathrm{op}}\bigg)\big\|\overline{\Theta}^{0}\boldsymbol{x}\big\|_2$$
$$\leq \bigg(\prod_{k=0}^{j-2}\big\|\overline{\Theta}^{k}\big\|_{\mathrm{op}}\bigg)\|\boldsymbol{x}\|_2$$

for all $j \in \{2,\ldots,l\}$. Now, since $\|\overline{\Theta}^{k}\|_{\mathrm{op}} \leq \|\overline{\Theta}^{k}\|_{\mathrm{F}} \leq \|\overline{\Theta}^{k}\|_{2,1}$ and $\overline{\Theta} \in \overline{\mathcal{M}}_{2,1}$, we can deduce from the display that

$$\big\|S_{j-2}\overline{\boldsymbol{g}}_{\overline{\Theta}}[\boldsymbol{x}]\big\|_2 \leq \bigg(\prod_{k=0}^{j-2}\big\|\overline{\Theta}^{k}\big\|_{2,1}\bigg)\|\boldsymbol{x}\|_2\,.$$

This inequality is our *contraction property*.

By similar arguments, we get for every $\boldsymbol{z}_1, \boldsymbol{z}_2 \in \mathbb{R}^{p^j}$ that

$$\big\|S^{j}\overline{\boldsymbol{g}}_{\overline{\Theta}}[\boldsymbol{z}_1] - S^{j}\overline{\boldsymbol{g}}_{\overline{\Theta}}[\boldsymbol{z}_2]\big\|_2$$
$$= \big\|\boldsymbol{f}^{l}\big[\overline{\Theta}^{l-1}\cdots\boldsymbol{f}^{j}[\boldsymbol{z}_1]\big] - \boldsymbol{f}^{l}\big[\overline{\Theta}^{l-1}\cdots\boldsymbol{f}^{j}[\boldsymbol{z}_2]\big]\big\|_2$$
$$\leq \big\|\overline{\Theta}^{l-1}\big[\boldsymbol{f}^{l-1}\cdots\boldsymbol{f}^{j}[\boldsymbol{z}_1]\big] - \overline{\Theta}^{l-1}\big[\boldsymbol{f}^{l-1}\cdots\boldsymbol{f}^{j}[\boldsymbol{z}_2]\big]\big\|_2$$
$$\leq \big\|\overline{\Theta}^{l-1}\big\|_{\mathrm{op}}\big\|\boldsymbol{f}^{l-1}\big[\cdots\boldsymbol{f}^{j}[\boldsymbol{z}_1]\big] - \boldsymbol{f}^{l-1}\big[\cdots\boldsymbol{f}^{j}[\boldsymbol{z}_2]\big]\big\|_2$$
$$\leq \cdots$$
$$\leq \bigg(\prod_{k=j}^{l-1}\big\|\overline{\Theta}^{k}\big\|_{\mathrm{op}}\bigg)\|\boldsymbol{z}_1 - \boldsymbol{z}_2\|_2$$

for $j \in \{1,\ldots,l\}$, where $\prod_{k=l}^{l-1}\|\overline{\Theta}^{k}\|_{\mathrm{op}} := 1$. Hence, similarly as above,

$$\big\|S^{j}\overline{\boldsymbol{g}}_{\overline{\Theta}}[\boldsymbol{z}_1] - S^{j}\overline{\boldsymbol{g}}_{\overline{\Theta}}[\boldsymbol{z}_2]\big\|_2 \leq \bigg(\prod_{k=j}^{l-1}\big\|\overline{\Theta}^{k}\big\|_{2,1}\bigg)\|\boldsymbol{z}_1 - \boldsymbol{z}_2\|_2\,.$$

This inequality is our *Lipschitz property*.

We now use the contraction and Lipschitz properties of the subnetworks to derive a Lipschitz result for the entire network. We consider two networks $\overline{\boldsymbol{g}}_{\overline{\Theta}}$ and $\overline{\boldsymbol{g}}_{\overline{\Gamma}}$ with parameters $\overline{\Theta} = (\overline{\Theta}^{l-1},\ldots,\overline{\Theta}^{0}) \in \overline{\mathcal{M}}_{2,1}$ and $\overline{\Gamma} = (\overline{\Gamma}^{l-1},\ldots,\overline{\Gamma}^{0}) \in \overline{\mathcal{M}}_{2,1}$, respectively. Our above-derived splitting argument applied with $j = 1$ and $j = l$, respectively, yields

$$\big\|\overline{\boldsymbol{g}}_{\overline{\Theta}}[\boldsymbol{x}] - \overline{\boldsymbol{g}}_{\overline{\Gamma}}[\boldsymbol{x}]\big\|_2 = \big\|S^{1}\overline{\boldsymbol{g}}_{\overline{\Theta}}\big[S_0\overline{\boldsymbol{g}}_{\overline{\Theta}}[\boldsymbol{x}]\big] - S^{l}\overline{\boldsymbol{g}}_{\overline{\Gamma}}\big[S_{l-1}\overline{\boldsymbol{g}}_{\overline{\Gamma}}[\boldsymbol{x}]\big]\big\|_2\,.$$

Elementary algebra and the fact that $S^{j-1}\overline{\boldsymbol{g}}_{\overline{\Theta}}[S_{j-2}\overline{\boldsymbol{g}}_{\overline{\Gamma}}[\boldsymbol{x}]] = S^{j}\overline{\boldsymbol{g}}_{\overline{\Theta}}[\overline{\Theta}^{j-1}\boldsymbol{f}^{j-1}[S_{j-2}\overline{\boldsymbol{g}}_{\overline{\Gamma}}[\boldsymbol{x}]]$ for $j \in \{2,\ldots,l\}$ then allow us to derive

$$\big\|\overline{\boldsymbol{g}}_{\overline{\Theta}}[\boldsymbol{x}] - \overline{\boldsymbol{g}}_{\overline{\Gamma}}[\boldsymbol{x}]\big\|_2$$
$$= \bigg\|S^{1}\overline{\boldsymbol{g}}_{\overline{\Theta}}\big[S_0\overline{\boldsymbol{g}}_{\overline{\Theta}}[\boldsymbol{x}]\big] - \sum_{j=1}^{l}\Big(S^{j}\overline{\boldsymbol{g}}_{\overline{\Theta}}\big[S_{j-1}\overline{\boldsymbol{g}}_{\overline{\Gamma}}[\boldsymbol{x}]\big] - S^{j}\overline{\boldsymbol{g}}_{\overline{\Theta}}\big[S_{j-1}\overline{\boldsymbol{g}}_{\overline{\Gamma}}[\boldsymbol{x}]\big]\Big) - S^{l}\overline{\boldsymbol{g}}_{\overline{\Gamma}}\big[S_{l-1}\overline{\boldsymbol{g}}_{\overline{\Gamma}}[\boldsymbol{x}]\big]\bigg\|_2$$
$$= \bigg\|S^{1}\overline{\boldsymbol{g}}_{\overline{\Theta}}\big[S_0\overline{\boldsymbol{g}}_{\overline{\Theta}}[\boldsymbol{x}]\big] - S^{1}\overline{\boldsymbol{g}}_{\overline{\Theta}}\big[S_0\overline{\boldsymbol{g}}_{\overline{\Gamma}}[\boldsymbol{x}]\big]$$
$$\quad - \sum_{j=2}^{l}\Big(S^{j}\overline{\boldsymbol{g}}_{\overline{\Theta}}\big[S_{j-1}\overline{\boldsymbol{g}}_{\overline{\Gamma}}[\boldsymbol{x}]\big] - S^{j-1}\overline{\boldsymbol{g}}_{\overline{\Theta}}\big[S_{j-2}\overline{\boldsymbol{g}}_{\overline{\Gamma}}[\boldsymbol{x}]\big]\Big)$$
$$\quad + S^{l}\overline{\boldsymbol{g}}_{\overline{\Theta}}\big[S_{l-1}\overline{\boldsymbol{g}}_{\overline{\Gamma}}[\boldsymbol{x}]\big] - S^{l}\overline{\boldsymbol{g}}_{\overline{\Gamma}}\big[S_{l-1}\overline{\boldsymbol{g}}_{\overline{\Gamma}}[\boldsymbol{x}]\big]\bigg\|_2$$

$$
\begin{aligned}
&= \Big\| S^1 \overline{\boldsymbol{g}}_{\overline{\boldsymbol{\Theta}}} \big[ S_0 \overline{\boldsymbol{g}}_{\overline{\boldsymbol{\Theta}}}[\boldsymbol{x}] \big] - S^1 \overline{\boldsymbol{g}}_{\overline{\boldsymbol{\Theta}}} \big[ S_0 \overline{\boldsymbol{g}}_{\overline{\boldsymbol{\Gamma}}}[\boldsymbol{x}] \big] \\
&\quad - \sum_{j=2}^{l} \Big( S^j \overline{\boldsymbol{g}}_{\overline{\boldsymbol{\Theta}}} \big[ S_{j-1} \overline{\boldsymbol{g}}_{\overline{\boldsymbol{\Gamma}}}[\boldsymbol{x}] \big] - S^j \overline{\boldsymbol{g}}_{\overline{\boldsymbol{\Theta}}} \big[ \overline{\Theta}^{j-1} \boldsymbol{f}^{j-1} \big[ S_{j-2} \overline{\boldsymbol{g}}_{\overline{\boldsymbol{\Gamma}}}[\boldsymbol{x}] \big] \big] \Big) \\
&\quad + S^l \overline{\boldsymbol{g}}_{\overline{\boldsymbol{\Theta}}} \big[ S_{l-1} \overline{\boldsymbol{g}}_{\overline{\boldsymbol{\Gamma}}}[\boldsymbol{x}] \big] - S^l \overline{\boldsymbol{g}}_{\overline{\boldsymbol{\Gamma}}} \big[ S_{l-1} \overline{\boldsymbol{g}}_{\overline{\boldsymbol{\Gamma}}}[\boldsymbol{x}] \big] \Big\|_2 \\
&\leq \big\| S^1 \overline{\boldsymbol{g}}_{\overline{\boldsymbol{\Theta}}} \big[ S_0 \overline{\boldsymbol{g}}_{\overline{\boldsymbol{\Theta}}}[\boldsymbol{x}] \big] - S^1 \overline{\boldsymbol{g}}_{\overline{\boldsymbol{\Theta}}} \big[ S_0 \overline{\boldsymbol{g}}_{\overline{\boldsymbol{\Gamma}}}[\boldsymbol{x}] \big] \big\|_2 \\
&\quad + \sum_{j=2}^{l} \big\| S^j \overline{\boldsymbol{g}}_{\overline{\boldsymbol{\Theta}}} \big[ S_{j-1} \overline{\boldsymbol{g}}_{\overline{\boldsymbol{\Gamma}}}[\boldsymbol{x}] \big] - S^j \overline{\boldsymbol{g}}_{\overline{\boldsymbol{\Theta}}} \big[ \overline{\Theta}^{j-1} \boldsymbol{f}^{j-1} \big[ S_{j-2} \overline{\boldsymbol{g}}_{\overline{\boldsymbol{\Gamma}}}[\boldsymbol{x}] \big] \big] \big\|_2 \\
&\quad + \big\| S^l \overline{\boldsymbol{g}}_{\overline{\boldsymbol{\Theta}}} \big[ S_{l-1} \overline{\boldsymbol{g}}_{\overline{\boldsymbol{\Gamma}}}[\boldsymbol{x}] \big] - S^l \overline{\boldsymbol{g}}_{\overline{\boldsymbol{\Gamma}}} \big[ S_{l-1} \overline{\boldsymbol{g}}_{\overline{\boldsymbol{\Gamma}}}[\boldsymbol{x}] \big] \big\|_2 .
\end{aligned}
$$

We bound this further by using the above-derived Lipschitz property of the outer networks and the observation that $S^l \overline{\boldsymbol{g}}_{\overline{\boldsymbol{\Theta}}}[S_{l-1}\overline{\boldsymbol{g}}_{\overline{\boldsymbol{\Gamma}}}[\boldsymbol{x}]] = S^l \overline{\boldsymbol{g}}_{\overline{\boldsymbol{\Gamma}}}[S_{l-1}\overline{\boldsymbol{g}}_{\overline{\boldsymbol{\Gamma}}}[\boldsymbol{x}]]$:

$$
\begin{aligned}
\big\| \overline{\boldsymbol{g}}_{\overline{\boldsymbol{\Theta}}}[\boldsymbol{x}] - \overline{\boldsymbol{g}}_{\overline{\boldsymbol{\Gamma}}}[\boldsymbol{x}] \big\|_2 &\leq \bigg( \prod_{k=1}^{l-1} \|\overline{\Theta}^k\|_{2,1} \bigg) \big\| S_0 \overline{\boldsymbol{g}}_{\overline{\boldsymbol{\Theta}}}[\boldsymbol{x}] - S_0 \overline{\boldsymbol{g}}_{\overline{\boldsymbol{\Gamma}}}[\boldsymbol{x}] \big\|_2 \\
&\quad + \sum_{j=2}^{l} \bigg( \prod_{k=j}^{l-1} \|\overline{\Theta}^k\|_{2,1} \bigg) \big\| S_{j-1} \overline{\boldsymbol{g}}_{\overline{\boldsymbol{\Gamma}}}[\boldsymbol{x}] - \overline{\Theta}^{j-1} \boldsymbol{f}^{j-1} \big[ S_{j-2} \overline{\boldsymbol{g}}_{\overline{\boldsymbol{\Gamma}}}[\boldsymbol{x}] \big] \big\|_2 ,
\end{aligned}
$$

which is by the definition of the inner networks equivalent to

$$
\begin{aligned}
\big\| \overline{\boldsymbol{g}}_{\overline{\boldsymbol{\Theta}}}[\boldsymbol{x}] - \overline{\boldsymbol{g}}_{\overline{\boldsymbol{\Gamma}}}[\boldsymbol{x}] \big\|_2 &\leq \bigg( \prod_{k=1}^{l-1} \|\overline{\Theta}^k\|_{2,1} \bigg) \|\overline{\Theta}^0 \boldsymbol{x} - \overline{\Gamma}^0 \boldsymbol{x}\|_2 \\
&\quad + \sum_{j=2}^{l} \bigg( \prod_{k=j}^{l-1} \|\overline{\Theta}^k\|_{2,1} \bigg) \big\| \overline{\Gamma}^{j-1} \boldsymbol{f}^{j-1} \big[ S_{j-2} \overline{\boldsymbol{g}}_{\overline{\boldsymbol{\Gamma}}}[\boldsymbol{x}] \big] - \overline{\Theta}^{j-1} \boldsymbol{f}^{j-1} \big[ S_{j-2} \overline{\boldsymbol{g}}_{\overline{\boldsymbol{\Gamma}}}[\boldsymbol{x}] \big] \big\|_2 .
\end{aligned}
$$

Using the properties of the operator norm, we can deduce from this inequality that

$$
\begin{aligned}
\big\| \overline{\boldsymbol{g}}_{\overline{\boldsymbol{\Theta}}}[\boldsymbol{x}] - \overline{\boldsymbol{g}}_{\overline{\boldsymbol{\Gamma}}}[\boldsymbol{x}] \big\|_2 &\leq \bigg( \prod_{k=1}^{l-1} \|\overline{\Theta}^k\|_{2,1} \bigg) \|\overline{\Theta}^0 - \overline{\Gamma}^0\|_{\mathrm{op}} \|\boldsymbol{x}\|_2 \\
&\quad + \sum_{j=2}^{l} \bigg( \prod_{k=j}^{l-1} \|\overline{\Theta}^k\|_{2,1} \bigg) \|\overline{\Gamma}^{j-1} - \overline{\Theta}^{j-1}\|_{\mathrm{op}} \big\| \boldsymbol{f}^{j-1} \big[ S_{j-2} \overline{\boldsymbol{g}}_{\overline{\boldsymbol{\Gamma}}}[\boldsymbol{x}] \big] \big\|_2 .
\end{aligned}
$$

Invoking the mentioned conditions on the activation functions and the contraction property for the inner subnetworks then yields

$$
\begin{aligned}
\big\| \overline{\boldsymbol{g}}_{\overline{\boldsymbol{\Theta}}}[\boldsymbol{x}] - \overline{\boldsymbol{g}}_{\overline{\boldsymbol{\Gamma}}}[\boldsymbol{x}] \big\|_2 &\leq \bigg( \max_{v \in \{0,\ldots,l-1\}} \prod_{\substack{k \in \{0,\ldots,l-1\} \\ k \neq v}} \max\{ \|\overline{\Theta}^k\|_{2,1}, \|\overline{\Gamma}^k\|_{2,1} \} \bigg) \bigg( \sum_{j=0}^{l-1} \|\overline{\Gamma}^j - \overline{\Theta}^j\|_{\mathrm{op}} \bigg) \|\boldsymbol{x}\|_2 \\
&\leq \sqrt{l} \|\boldsymbol{x}\|_2 \|\overline{\boldsymbol{\Theta}} - \overline{\boldsymbol{\Gamma}}\|_{\mathrm{F}} .
\end{aligned}
$$

The proof for the connection-sparse case is almost the same. The main difference is that one needs to use the $\|\cdot\|_\infty$- and $\|\cdot\|_1$-norms (rather than the $\|\cdot\|_2$- and $\|\cdot\|_{\mathrm{op}}$-norms) and the inequality $\|A\boldsymbol{b}\|_\infty \leq \|A\|_1 \|\boldsymbol{b}\|_\infty$ (rather than the inequality $\|A\boldsymbol{b}\|_2 \leq \|A\|_{\mathrm{op}} \|\boldsymbol{b}\|_2$) to establish suitable contraction and Lipschitz properties. $\qquad \square$

## A.2 ENTROPY BOUND

In this section, we establish bounds for the entropies of $\overline{\mathcal{M}}_1$ and $\overline{\mathcal{M}}_{2,1}$. The distance between two networks $\overline{\boldsymbol{g}}_{\overline{\boldsymbol{\Theta}}}$ and $\overline{\boldsymbol{g}}_{\overline{\boldsymbol{\Gamma}}}$ is defined as $\mathrm{dist}[\overline{\boldsymbol{g}}_{\overline{\boldsymbol{\Theta}}}, \overline{\boldsymbol{g}}_{\overline{\boldsymbol{\Gamma}}}] := \sqrt{\sum_{i=1}^{n} \|\overline{\boldsymbol{g}}_{\overline{\boldsymbol{\Theta}}}[\boldsymbol{x}_i] - \overline{\boldsymbol{g}}_{\overline{\boldsymbol{\Gamma}}}[\boldsymbol{x}_i]\|_\infty^2 / n}$. Given this distance function and a radius $t \in (0, \infty)$, the metric entropy of a nonempty set $\mathcal{A} \subset \{ \overline{\boldsymbol{\Theta}} = (\overline{\Theta}^{l-1}, \ldots, \overline{\Theta}^0) : \overline{\Theta}^j \in \mathbb{R}^{p^{j+1} \times p^j} \}$ is denoted by $H[t, \mathcal{A}]$. We then get the following entropy bounds.

**Lemma 1** (Entropy Bounds). *In the framework of Sections 2 and 3, it holds for a constant $c_H \in (0, \infty)$ and every $t \in (0, \infty)$ that*

$$H[t, \overline{\mathcal{M}}_1] \leq c_H \left\lceil \frac{(\overline{v}_\infty)^2 l}{t^2} \right\rceil \log \left[ \frac{\overline{p}t^2}{(\overline{v}_\infty)^2 l} + 2 \right]$$

*and*

$$H[t, \overline{\mathcal{M}}_{2,1}] \leq c_H \left\lceil \frac{(\overline{v}_\infty)^2 lp}{t^2} \right\rceil \log \left[ \frac{\overline{p}t^2}{(\overline{v}_\infty)^2 l} + 2 \right].$$

*Proof of Lemma 1.* The first bound can be derived by combining established deterministic and randomization arguments (Carl, 1985);(Lederer, 2010, Proof of Theorem 1.1);(Taheri et al., 2020, Proposition 3).

For the second bound, observe that

$$\|\Theta^j\|_1 = \sum_{i=1}^{p^{j+1}} \sum_{k=1}^{p^j} |(\Theta^j)_{ik}| \leq \sqrt{p^{j+1}} \sum_{k=1}^{p^j} \sqrt{\sum_{i=1}^{p^{j+1}} |(\Theta^j)_{ik}|^2} = \sqrt{p^{j+1}} \|\Theta^j\|_{2,1} = \sqrt{\overline{p}} \|\Theta^j\|_{2,1}$$

for all $j \in \{0, \dots, l-1\}$ and $\Theta^j \in \mathbb{R}^{p^{j+1} \times p^j}$. We used in turn 1. the definition of the $\|\cdot\|_1$-norm on Page 2, 2. the linearity and interchangeability of finite sums and the inequality $\|a\|_1 \leq \sqrt{b}\|a\|_2$ for all $a \in \mathbb{R}^b$, 3. the definition of the $\|\cdot\|_{2,1}$-norm on Page 4, and 4. the definition of the width $\underline{p}$ on Page 2. Hence, $\overline{\mathcal{M}}_{2,1} \subset \sqrt{\overline{p}} \overline{\mathcal{M}}_1$. A bound for the entropies of $\overline{\mathcal{M}}_{2,1}$ can, therefore, be derived from the first bound by replacing the radii $t$ on the right-hand side by $t/\sqrt{\overline{p}}$. $\qquad\square$

### A.3 PROOF OF THEOREM 1

In this section, we state a proof for Theorem 1. The proof is inspired by derivations in high-dimensional statistics—see, for example, (Zhuang & Lederer, 2018) and references therein.

*Proof of Theorem 1.* The main idea of the proof is to contrast the estimators' objective functions evaluated at their minima with the estimators' objective functions at other points. Our first step is to derive what we call a *basic inequality*. By the definition of the estimator in (6), it holds for every $\Theta \in \mathcal{M}_{2,1}$ that

$$\sum_{i=1}^n \left\| y_i - g_{\widehat{\Theta}}[x_i] \right\|_2^2 + r_{\text{node}} \|\widehat{\Theta}^l\|_{2,1} \leq \sum_{i=1}^n \left\| y_i - g_\Theta[x_i] \right\|_2^2 + r_{\text{node}} \|\Theta^l\|_{2,1},$$

where we use the shorthand $\widehat{\Theta} := \widehat{\Theta}_{\text{node}}$. We then invoke the model in (1) to rewrite this inequality as

$$\sum_{i=1}^n \left\| g_*[x_i] + u_i - g_{\widehat{\Theta}}[x_i] \right\|_2^2 + r_{\text{node}} \|\widehat{\Theta}^l\|_{2,1} \leq \sum_{i=1}^n \left\| g_*[x_i] + u_i - g_\Theta[x_i] \right\|_2^2 + r_{\text{node}} \|\Theta^l\|_{2,1}.$$

Expanding the squared terms and rearranging the inequality then yields

$$\sum_{i=1}^n \left\| g_*[x_i] - g_{\widehat{\Theta}}[x_i] \right\|_2^2 \leq \sum_{i=1}^n \left\| g_*[x_i] - g_\Theta[x_i] \right\|_2^2$$

$$+ 2\sum_{i=1}^n \left( g_{\widehat{\Theta}}[x_i] \right)^\top u_i - 2\sum_{i=1}^n \left( g_\Theta[x_i] \right)^\top u_i + r_{\text{node}} \|\Theta^l\|_{2,1} - r_{\text{node}} \|\widehat{\Theta}^l\|_{2,1}.$$

This is our basic inequality.

In the remainder of the proof, we need to bound the first two terms in the last line of the basic inequality. We call these terms the *empirical process terms*. Using the reformulation of the networks in (7), we can write the empirical process term of a general parameter $\Gamma \in \mathcal{M}_{2,1}$ according to

$$2\sum_{i=1}^n \left( g_\Gamma[x_i] \right)^\top u_i = 2\sum_{i=1}^n \left( \Gamma^l \overline{g}_{\overline{\Gamma}}[x_i] \right)^\top u_i$$

with $\overline{\Gamma} \in \overline{\mathcal{M}}_{2,1}$. Using the 1. the properties of transpositions, 2. the definition of the trace function, 3. the cyclic property of the trace function, and 4. the linearity of the trace function yields further

$$
\begin{aligned}
2\sum_{i=1}^{n}\left(\boldsymbol{g}_{\Gamma}[\boldsymbol{x}_i]\right)^{\top}\boldsymbol{u}_i &= 2\sum_{i=1}^{n}\left(\overline{\boldsymbol{g}}_{\overline{\Gamma}}[\boldsymbol{x}_i]\right)^{\top}(\Gamma^l)^{\top}\boldsymbol{u}_i \\
&= 2\sum_{i=1}^{n}\operatorname{trace}\left[\left(\overline{\boldsymbol{g}}_{\overline{\Gamma}}[\boldsymbol{x}_i]\right)^{\top}(\Gamma^l)^{\top}\boldsymbol{u}_i\right] \\
&= 2\sum_{i=1}^{n}\operatorname{trace}\left[\boldsymbol{u}_i\left(\overline{\boldsymbol{g}}_{\overline{\Gamma}}[\boldsymbol{x}_i]\right)^{\top}(\Gamma^l)^{\top}\right] \\
&= 2\operatorname{trace}\left[\left(\sum_{i=1}^{n}\boldsymbol{u}_i\left(\overline{\boldsymbol{g}}_{\overline{\Gamma}}[\boldsymbol{x}_i]\right)^{\top}\right)(\Gamma^l)^{\top}\right].
\end{aligned}
$$

Now, 1. denoting the column-vector that corresponds to the $k$th column of a matrix $A$ by $A_{\bullet k}$, 2. using Hölder's inequality, 3. using Hölder's inequality again, and 4. again Hölder's inequality and our definitions of the elementwise $\ell_\infty$-and $\ell_1$-norms, we find

$$
\begin{aligned}
2\sum_{i=1}^{n}\left(\boldsymbol{g}_{\Gamma}[\boldsymbol{x}_i]\right)^{\top}\boldsymbol{u}_i &= 2\sum_{k=1}^{p^l}\left\langle\left(\sum_{i=1}^{n}\boldsymbol{u}_i\left(\overline{\boldsymbol{g}}_{\overline{\Gamma}}[\boldsymbol{x}_i]\right)^{\top}\right)_{\bullet k},(\Gamma^l)_{\bullet k}\right\rangle \\
&\leq 2\sum_{k=1}^{p^l}\left\|\left(\sum_{i=1}^{n}\boldsymbol{u}_i\left(\overline{\boldsymbol{g}}_{\overline{\Gamma}}[\boldsymbol{x}_i]\right)^{\top}\right)_{\bullet k}\right\|_2\left\|(\Gamma^l)_{\bullet k}\right\|_2 \\
&\leq 2\max_{k\in\{1,\dots,p^l\}}\left\|\left(\sum_{i=1}^{n}\boldsymbol{u}_i\left(\overline{\boldsymbol{g}}_{\overline{\Gamma}}[\boldsymbol{x}_i]\right)^{\top}\right)_{\bullet k}\right\|_2\sum_{k=1}^{p^l}\left\|(\Gamma^l)_{\bullet k}\right\|_2 \\
&\leq 2\sqrt{m}\left\|\sum_{i=1}^{n}\boldsymbol{u}_i\left(\overline{\boldsymbol{g}}_{\overline{\Gamma}}[\boldsymbol{x}_i]\right)^{\top}\right\|_\infty\left\|\!\left|\Gamma^l\right|\!\right\|_{2,1},
\end{aligned}
$$

which implies in view of the definition of the effective noise in (8)

$$
2\sum_{i=1}^{n}\left(\boldsymbol{g}_{\Gamma}[\boldsymbol{x}_i]\right)^{\top}\boldsymbol{u}_i \leq r^*_{\text{node}}\left\|\!\left|\Gamma^l\right|\!\right\|_{2,1}.
$$

This inequality is our bound on the empirical process terms.

We can combine the bound on the empiricial process term and the basic inequality to find

$$
\begin{aligned}
\sum_{i=1}^{n}\left\|\boldsymbol{g}_*[\boldsymbol{x}_i]-\boldsymbol{g}_{\widehat{\Theta}}[\boldsymbol{x}_i]\right\|_2^2 \leq &\sum_{i=1}^{n}\left\|\boldsymbol{g}_*[\boldsymbol{x}_i]-\boldsymbol{g}_{\Theta}[\boldsymbol{x}_i]\right\|_2^2 \\
&+ r^*_{\text{node}}\|\widehat{\Theta}^l\|_{2,1}+r^*_{\text{node}}\|\Theta^l\|_{2,1}+r_{\text{node}}\|\Theta^l\|_{2,1}-r_{\text{node}}\|\widehat{\Theta}^l\|_{2,1}.
\end{aligned}
$$

Using then the assumption $r_{\text{node}} \geq r^*_{\text{node}}$ yields

$$
\sum_{i=1}^{n}\left\|\boldsymbol{g}_*[\boldsymbol{x}_i]-\boldsymbol{g}_{\widehat{\Theta}}[\boldsymbol{x}_i]\right\|_2^2 \leq \sum_{i=1}^{n}\left\|\boldsymbol{g}_*[\boldsymbol{x}_i]-\boldsymbol{g}_{\Theta}[\boldsymbol{x}_i]\right\|_2^2 + 2r_{\text{node}}\|\Theta^l\|_{2,1}.
$$

Multiplying both sides by $1/n$ and taking the infimum over $\Theta \in \mathcal{M}_{2,1}$ on the right-hand side then gives

$$
\frac{1}{n}\sum_{i=1}^{n}\left\|\boldsymbol{g}_*[\boldsymbol{x}_i]-\boldsymbol{g}_{\widehat{\Theta}}[\boldsymbol{x}_i]\right\|_2^2 \leq \inf_{\Theta\in\mathcal{M}_{2,1}}\left\{\frac{1}{n}\sum_{i=1}^{n}\left\|\boldsymbol{g}_*[\boldsymbol{x}_i]-\boldsymbol{g}_{\Theta}[\boldsymbol{x}_i]\right\|_2^2 + \frac{2r_{\text{node}}}{n}\|\Theta^l\|_{2,1}\right\}.
$$

Invoking the definition of the prediction error on Page 4 gives the desired result.

The proof for the connection-sparse estimator is virtually the same. $\qquad\square$

### A.4 PROOF OF PROPOSITION 1

In this section, we give a short proof of Proposition 1.

*Proof of Proposition 1.* Verify the fact that if the all-zeros parameter is neither a solution of (3) nor of (5), all solutions $\widehat{\boldsymbol{\Theta}}_{\text{con}}$ and $\widetilde{\boldsymbol{\Theta}}_{\text{con}}$ of (3) and (5), respectively, satisfy $(\widehat{\Theta}_{\text{con}})^j, (\widetilde{\Theta}_{\text{con}})^j \neq \mathbf{0}_{p^{j+1} \times p^j}$ for all $j \in \{0, \ldots, l\}$.

It then follows from the assumed nonnegative homogeneity, $r_{\text{con}} > 0$, and the definition of the estimator in (3) that $\|(\widehat{\Theta}_{\text{con}})^0\|_1, \ldots, \|(\widehat{\Theta}_{\text{con}})^{l-1}\|_1 = 1$ for all solutions $\widehat{\boldsymbol{\Theta}}_{\text{con}}$.

Given a solution $\widetilde{\boldsymbol{\Theta}}_{\text{con}}$ of (5), define $a := \|(\widetilde{\Theta}_{\text{con}})^0\|_1/(l+1) + \cdots + \|(\widetilde{\Theta}_{\text{con}})^l\|_1/(l+1)$ and verify the fact that $\boldsymbol{\Gamma} \in \mathcal{M}$ with $\Gamma^0 := a(\widetilde{\Theta}_{\text{con}})^0/\|(\widetilde{\Theta}_{\text{con}})^0\|_1, \Gamma^1 := a(\widetilde{\Theta}_{\text{con}})^1/\|(\widetilde{\Theta}_{\text{con}})^1\|_1, \ldots$ has the same value in the objective function as $\widetilde{\boldsymbol{\Theta}}_{\text{con}}$.

$\square$

### A.5 PROOF OF PROPOSITION 2

In this section, we establish a proof of Proposition 2. The key tools are the Lipschitz property of Proposition 4 and the entropy bounds of Lemma 1.

*Proof of Proposition 2.* The main idea is to rewrite the event under consideration in a form that is amenable to known tail bounds for suprema of empirical processes with subgaussian random variables.

The connection-sparse bound follows from

$$
\begin{aligned}
&P\left\{ r^*_{\text{con}} \geq c\overline{v}_\infty \sqrt{nl\big(\log[2mn\overline{p}]\big)^3} \right\} \\
&= P\left\{ 2 \sup_{\overline{\boldsymbol{\Psi}} \in \overline{\mathcal{M}}_1} \left\| \sum_{i=1}^n \boldsymbol{u}_i \big(\overline{\boldsymbol{g}}_{\overline{\boldsymbol{\Psi}}}[\boldsymbol{x}_i]\big)^\top \right\|_\infty \geq c\overline{v}_\infty \sqrt{nl\big(\log[2mn\overline{p}]\big)^3} \right\} \\
&\leq mp^l \max_{\substack{j \in \{1,\ldots,m\} \\ k \in \{1,\ldots,p^l\}}} P\left\{ 2 \sup_{\overline{\boldsymbol{\Psi}} \in \overline{\mathcal{M}}_1} \left| \left(\sum_{i=1}^n \boldsymbol{u}_i \big(\overline{\boldsymbol{g}}_{\overline{\boldsymbol{\Psi}}}[\boldsymbol{x}_i]\big)^\top\right)_{jk} \right| \geq c\overline{v}_\infty \sqrt{nl\big(\log[2mn\overline{p}]\big)^3} \right\} \\
&\leq mp^l \cdot \frac{1}{mn\overline{p}} \\
&\leq \frac{1}{n},
\end{aligned}
$$

where we use in turn 1. the definition of $r^*_{\text{con}}$ in (8), 2. the union bound, 3. van de Geer (2000, Corollary 8.3) and our Proposition 4 and Lemma 1, and 4. the inequality $p^l \leq \overline{p} = \sum_{j=0}^l p^{j+1}p^j$ and consolidating the factors. The key concept underlying van de Geer (2000, Corollary 8.3 on Page 128) is chaining (van der Vaart & Wellner, 1996, Page 90).

The same considerations also apply to the node-sparse case, but we get an additional factor $\sqrt{m}$ from the definition of the effective noise in (8) and a factor $\sqrt{\overline{p}}$ from the entropy bound in Lemma 1. The differences between the bounds for the connection- and node-sparse cases in terms of $\overline{v}_\infty$ vs. $\overline{v}_2$ stem from the different Lipschitz constants in Proposition 4. $\square$

### A.6 PROOF OF PROPOSITION 3

*Proof of Proposition 3.* The proof is based on standard empirical-process theory, including contraction and symmetrization arguments.

Using basic algebra and measure theory, it is easy to show that

$$\text{risk}[\widehat{\boldsymbol{\Theta}}_{\text{con}}] \leq (1+b)\,\text{risk}[\boldsymbol{\Theta}^*] + c_b\,\text{err}[\widehat{\boldsymbol{\Theta}}_{\text{con}}]$$
$$+ c_b \left| \frac{1}{n} \sum_{i=1}^{n} \left( \left\| \boldsymbol{g}_*[\boldsymbol{x}_i] - \boldsymbol{g}_{\widehat{\boldsymbol{\Theta}}_{\text{con}}}[\boldsymbol{x}_i] \right\|_2^2 - E \left\| \boldsymbol{g}_*[\boldsymbol{x}_i] - \boldsymbol{g}_{\widehat{\boldsymbol{\Theta}}_{\text{con}}}[\boldsymbol{x}_i] \right\|_2^2 \right) \right|$$

for a constant $c_b \in (0, \infty)$ that depends only on $b$. The first term in this bound is the minimal risk as stated in the proposition, and the second term can be bounded by Corollary 1 and Proposition 2. Hence, it remains to bound the third term.

In view of the law of large numbers, it is reasonable to hope for the third term to be small. But to make this precise, we have to keep in mind that the estimator itself depends on the input vectors. We, therefore, need to prepare the third term for the application of a uniform version of the law of large numbers. Using standard contraction arguments—see (Boucheron et al., 2013, Chapter 11.3), for example—and Hölder's inequality, we can bound the third term by bounding

$$\max\{\|(\boldsymbol{\Theta}^*)^l\|_1, \|(\widehat{\boldsymbol{\Theta}}_{\text{con}})^l\|_1\} \sup_{\overline{\boldsymbol{\Theta}} \in \overline{\mathcal{M}}_1} \left\| \sum_{i=1}^{n} \left( \overline{\boldsymbol{g}}_{\overline{\boldsymbol{\Theta}}^*}[\boldsymbol{x}_i] - \overline{\boldsymbol{g}}_{\overline{\boldsymbol{\Theta}}}[\boldsymbol{x}_i] - E[\overline{\boldsymbol{g}}_{\overline{\boldsymbol{\Theta}}^*}[\boldsymbol{x}_i] - \overline{\boldsymbol{g}}_{\overline{\boldsymbol{\Theta}}}[\boldsymbol{x}_i]] \right) \right\|_\infty^2 ,$$

which removes the dependence on the estimator $\widehat{\boldsymbol{\Theta}}_{\text{con}}$ up to the leading factor. To see that we can also neglect that factor, verify (see Proposition 2 and the proof of Theorem 1) that $\|(\widehat{\boldsymbol{\Theta}}_{\text{con}})^l\|_1 \leq 2\|(\boldsymbol{\Theta}^*)^l\|_1$ with high probability as long as $r_{\text{con}}^* \geq c\overline{v}_\infty \sqrt{nl(\log[2mn\overline{p}])^3}$ with $c$ large enough. Consequently, we just need to consider the quantity

$$\sup_{\overline{\boldsymbol{\Theta}} \in \overline{\mathcal{M}}_1} \left\| \sum_{i=1}^{n} \left( \overline{\boldsymbol{g}}_{\overline{\boldsymbol{\Theta}}^*}[\boldsymbol{x}_i] - \overline{\boldsymbol{g}}_{\overline{\boldsymbol{\Theta}}}[\boldsymbol{x}_i] - E[\overline{\boldsymbol{g}}_{\overline{\boldsymbol{\Theta}}^*}[\boldsymbol{x}_i] - \overline{\boldsymbol{g}}_{\overline{\boldsymbol{\Theta}}}[\boldsymbol{x}_i]] \right) \right\|_\infty^2$$

in the following.

The last step is to bring this term in a form that is amenable to our earlier proofs. Using standard symmetrization arguments—see van der Vaart & Wellner (1996, Chapter 2.3), for example)—we can bound this quantity by bounding

$$\sup_{\overline{\boldsymbol{\Theta}} \in \overline{\mathcal{M}}_1} \left\| \sum_{i=1}^{n} k_i \left( \overline{\boldsymbol{g}}_{\overline{\boldsymbol{\Theta}}^*}[\boldsymbol{x}_i] - \overline{\boldsymbol{g}}_{\overline{\boldsymbol{\Theta}}}[\boldsymbol{x}_i] \right) \right\|_\infty^2 ,$$

where $k_1, \ldots, k_n$ are i.i.d. Rademacher random variables. But even though $k_1, \ldots, k_n$ are i.i.d. Rademacher random variables, we do not resort to Rademacher complexities; instead, we use that Rademacher random variables are subgaussian, so that we can then proceed similarly as in the proof of Proposition 2.

The node-sparse case can be treated along the same lines. $\qquad\square$

## A.7 EXTENSIONS

Our proof approach disentangles the specifics of the objective function (proof of Theorem 1), of the network structure (proof of Proposition 4), and of the stochastic terms (proofs of Lemma 1 and Proposition 2). This feature allows one to generalize and extend the results of this paper in straight-forward ways. For example, extensions to different noise distributions only need a corresponding version of Proposition 2—with everything else unchanged. One could envision, for example, using concentration inequalities for heavy-tailed distributions such as in Lederer & van de Geer (2014). Extensions to different loss functions, to give another example, can be established by adjusting Theorem 1 accordingly. This can be done, for example, by invoking ideas from specialized literature on high-dimensional logistic regression such as Li & Lederer (2019). We avoid going into further details to avoid digression; the key message is that the flexibility of the proofs is yet another advantage of our approach.

