# OpenReview forum: "Connection- and Node-Sparse Deep Learning: Statistical Guarantees"
_ICLR.cc/2021/Conference — Reject_

### Official Review · AnonReviewer2 · 2020-10-17
**While the paper studies the important problem of prediction accuracy of neural networks, it only provides bounds on the in-sample error and not the generalization error.**

**Rating:** 5
**Confidence:** 3

**Review:**

# Summary
This paper studies the problem of estimating a vector valued regression function by neural networks. They provide a bound on the in-sample prediction error for a neural network estimator under two types of regularizations; one that induces connection sparsity and another that induces node sparsity. The in-sample error is bounded by the in sample error of an estimator computed in the noiseless case.

# Strengths
Statistical theory of the success of deep learning is an important problem currently. Moreover, studying the performance of neural network estimators under appropriate regularization provides insights into their practical success.

This paper provides some insights into the workings of different types of regularization commonly used.

The paper is well written and organized. The proofs are concise and clear.

# Weaknesses
However, my main concerns with this paper are as follows:
 -  The study of the in-sample prediction error and not the generalization error. A crucial problem in deep learning is to discover why neural networks that are over-parametrized have low generalization error. In fact, it has been suggested, for eg in Arora et. al. (	arXiv:1905.13655 ) that specific norm regularization may not lead to optimal generalization.
 -  The related work, specifically Neyshabur2015, study the generalization error or out of sample error so the comparison with their bound does not seem to be direct.
 - In general, it is not clarified how this work supersedes previous work. In particular while the dependence on the input vector is shown to be better, a concrete example of when such an improvement would manifest itself would be very helpful.
 - There seems to be other work that seems relevant but of this I am not certain for eg. Mhaskar and Poggio (arXiv:1802.06266). Also, a large amount of literature on bounds on generalization error in the case of classification i.e. Neyshabur 2017a (arXiv:1707.09564), Arora et. al. (arXiv:1802.05296).

---

### Official Review · AnonReviewer4 · 2020-10-28
**Regularized and Constrained Estimation are pretty similar**

**Rating:** 4
**Confidence:** 4

**Review:**

This paper studies mean-squared-error bounds for neural networks with small $\ell_1$-norm. The use of $\ell_1$-norm constraint is analogous to the use of LASSO in sparse linear regression. They give a "mean-squared-error" bound because they only analyze a fixed-design setting (where the goal is only to analyze the effect of noise) instead of the random-design setting which requires an additional analysis of generalization to fresh data.

They emphasize that they study the case where the returned network is regularized using an $\ell_1$-penalty (as in eqn (3)) instead of the case where the minimization is explicitly constrained to an $\ell_1$-ball.  As the authors say, regularization is more convenient to implement than constrained optimization in practice. They claim that this makes the problem they analyze quite different from analyzing the explicitly constrained version, because the minimization is technically over an unbounded class of functions. However, we know from the basic theory of Lagrange multipliers that the optimizer of the $\ell_1$-regularized version is also the optimal solution under some $\ell_1$-norm constraint. So conceptually, the difference between the regularized and constrained versions is not that important, although analyzing the regularized version creates some extra technical difficulties.

This paper makes some other overstated claims about the novelty of their result and analysis. The authors claim that their analysis is based off of some new techniques from high-dimensional statistics which have not appeared in the neural net generalization literature. However, their analysis is in fact almost the same as the usual generalization bounds. For example, the "generalized noise" (8) which they are concerned with is basically just the empirical Rademacher complexity of a single neuron of the last hidden layer (consider the case where the noise u is Rademacher).

Overall, I think this work does not provide much fresh insight into generalization bounds for neural networks, and I would tend towards rejection.

Minor notes:
- Older related work. The idea of analyzing generalization error based on $\ell_1$-norm (which associates with sparsity) can be traced back at least to the paper 'The Sample Complexity of Pattern Classification with Neural Networks: The Size of the Weights is More Important than the Size of the Network' by Bartlett '98.

---

### Official Review · AnonReviewer3 · 2020-10-28
**marginally above acceptance**

**Rating:** 6
**Confidence:** 3

**Review:**

##########################################################################

Summary:
This paper introduces theoretical statistical guarantees for different types of sparsity inducing regularization in regression type settings for deep layered feedforward networks, using techniques of high-dimensional statistics and empirical process theory.

##########################################################################

Pros:
1. The paper introduces techniques well-known in high dimensional linear regression for deriving guarantees in deep learning.
2. The paper introduces methods that are more elegant and simpler than previous methods.
3. The paper is well written, easy to understand and with clear definitions and connections between sections.
4. The comparisons with related work are well-addressed.

##########################################################################


Cons:
1. While the paper provides interesting theoretical results, it  would be interesting if the authors can show empirical consistency for some of their claims, for example, their conclusion that connection sparsity is suitable to handle wide networks, but node
sparsity is suitable only when complemented by connection sparsity or other strategies. Similarly, while the paper cannot provide ways to select regularization parameters, the dependence on m,n,p should be compared empirically.
2. The authors should clarify clearly the assumptions that makes their mathematical treatment feasible. The authors should also clarify the conditions under which their guarantees do not hold.
3. Are the authors' methods extendable to settings other than regression for example classification?



##########################################################################

Questions during rebuttal period:


Please address and clarify the cons above

---

### Decision · Program_Chairs · 2021-01-07
**Final Decision**

**Decision:**

Reject

**Comment:**

The paper considers the problem of using sparse coding to create better generalization in neural networks. The new generalization bound of the neural network only depends on the l1 norm of the weight, instead of the original \ell_2 version as in previous papers.


Well this direction is promising, the major concern about this work is that how they compare with existing generalization bounds empirically. There are definitely some hand-crafted instances where this bound excel, but the authors did not provide enough evidence that this bound would actually be better than others for neural networks trained in practice: For example, would adding a relatively large \ell_1 regularizer resulted in a drastic decrement in test accuracy? How does the bound compare with compression based approach such as VC dimension + weight pruning (since the weights are somewhat sparse, so the VC dimension is lower) -- One might argue that those pruning techniques do not have theoretical guarantees that they can work -- Well this technique does not have theoretical guarantees either (whether this objective can be minimized efficiently): The theorem, at least in the current form, seems to only apply to networks that are the "global optimals" of some non-convex training objective (MSE-loss involving a non-linear neural network +  ell_1 regularizer on its weights). It is also unclear whether such global optimals can be found efficiently in practice. At very least, the authors should devote some effort demonstrating the superiority of their bound empirically.